# Analyzing nested experimental designs—A user-friendly resampling method to determine experimental significance

**Rishikesh U. Kulkarni** [1] *, **Catherine L. Wang** [1], **Carolyn R. Bertozzi** [1,2,3] *

**1** Department of Chemistry, Stanford University, Stanford, California, United States of America, **2** Stanford ChEM-H, Stanford University, Stanford, California, United States of America, **3** Howard Hughes Medical Institute, Stanford University, Stanford, California, United States of America

* rishi@kulkarni.science (RUK); bertozzi@stanford.edu (CRB)

## Abstract

While hierarchical experimental designs are near-ubiquitous in neuroscience and biomedical research, researchers often do not take the structure of their datasets into account while performing statistical hypothesis tests. Resampling-based methods are a flexible strategy for performing these analyses but are difficult due to the lack of open-source software to automate test construction and execution. To address this, we present *Hierarch*, a Python package to perform hypothesis tests and compute confidence intervals on hierarchical experimental designs. Using a combination of permutation resampling and bootstrap aggregation, *Hierarch* can be used to perform hypothesis tests that maintain nominal Type I error rates and generate confidence intervals that maintain the nominal coverage probability without making distributional assumptions about the dataset of interest. *Hierarch* makes use of the Numba JIT compiler to reduce *p*-value computation times to under one second for typical datasets in biomedical research. *Hierarch* also enables researchers to construct user-defined resampling plans that take advantage of *Hierarch's* Numba-accelerated functions.

**Data Availability Statement:** The Hierarch source code is available at https://github.com/rishi-kulkarni/hierarch. The simulations performed in this manuscript, as well as the code to produce the

## Author summary

An important step in analyzing experimental data is quantifying uncertainty in the experimenter's conclusions. One mechanism for doing so is by using a statistical hypothesis test, which allows the experimenter to control what percentage of the time they make erroneous conclusions over the course of their career. Biological experimental designs often have hierarchical data-gathering schemes that traditional hypothesis tests are not well-suited for (for example, an experimenter may make measurements of several tissue samples that were collected from subjects who were given a treatment). While traditional tests can be adapted to hierarchical experimental designs, we propose a simple resampling-based hypothesis test that applies to a variety of experimental designs while maintaining control over error rate. In this manuscript, we describe *Hierarch*, the Python package that enables users to carry out this test and validate it under several conditions.

figures, are available as Jupyter Notebooks at
https://github.com/rishi-kulkarni/hierarch-
simulations.

**Funding:** This work was funded by the US National
Institutes of Health (Grant R01 CA227942 to CRB).
URL: https://grantome.com/grant/NIH/R01-
CA227942-18A1 The funders had no role in study
design, data collection and analysis, decision to
publish, or preparation of the manuscript.

**Competing interests:** The authors have declared
that no competing interests exist.

## Introduction

Typical experimental design in the life sciences produces hierarchical data (or clustered, nested, multilevel, etc.) [1–3]. For example, a researcher might image multiple fields of view from the same coverslip in an imaging experiment or record multiple trials from the same animal in a behavioral study (**Fig 1**). Despite the ubiquity of this type of experimental design, strategies for computing p-values for these experiments are hugely inconsistent in the literature. Common approaches range from "pseudoreplication" strategies that treat different fields of view as independent samples, to "summary statistic" approaches that aggregate the fields of view before performing a t-test or ANOVA [4–6]. These approaches can produce wildly different p-values on the same datasets because they do not consider the hierarchical nature of the experimental design. The p-value is commonly misunderstood to be a measure of the compatibility of the null hypothesis with the observed data; however, the p-value is more accurately defined as a measure of the compatibility of the entire statistical model (including ALL assumptions made by the hypothesis test) with the observed data [7]. If a researcher wishes to compute a useful p-value for a hierarchical dataset, the experimental design must factor into the statistical model in some manner.

One approach to analyzing hierarchical data is using a linear mixed model (or hierarchical model) [8,9]. Linear mixed models represent hierarchical data by being hierarchical themselves—the regression coefficients and intercept are themselves represented by another regression model. As flexible and powerful as they are, most studies employing linear mixed models involve very large numbers of clusters ($>$20), while studies in biomedical research typically have fewer than seven clusters and most often three to five [10,11]. Simulation studies have shown that linear mixed models fail to control Type I error (false positive) rates with such a small number of clusters, becoming conservative or liberal depending if the effect of interest is within-clusters or between-clusters [4,12]. Furthermore, the process of selecting parameters for a linear mixed model can be challenging–specifying the structure of a given data set is nontrivial, but failure to do so correctly completely invalidates the p-values computed by the model.

Ideally, researchers could analyze hierarchical data using a hypothesis test that incorporates data from every level of hierarchy, does not make any distributional assumptions about the dataset, and can be easily applied to a wide range of experimental designs. Randomization (or permutation) tests can be used to calculate p-values and confidence intervals while making only very weak assumptions about the nature of the data [13,14]. By accounting for each level of hierarchy in the resampling plan, a hierarchical randomization test can control false positive rates while achieving good statistical power. Furthermore, resampling-based tests can be "distribution-free" in the sense that they typically make weaker assumptions regarding the population distributions underlying the samples [15–17]. This has the added benefit of producing a p-value that does not depend on unverifiable assumptions about the data-generating process [18]. Despite the good properties of resampling-based tests, they come with a few drawbacks. One major drawback of this approach is that for a given dataset, the script executing the resampling plan is often bespoke and computationally intensive. Furthermore, incorrectly specifying the resampling plan can result in inflated Type I error rates the same way that choosing the wrong traditional hypothesis test can inflate Type I error rates. Nonetheless, biological experiments often are by their very nature hierarchical, and demand a statistical approach that keeps hierarchy in mind.

To address these challenges, we present *Hierarch*, a Python-based module for hierarchical hypothesis testing. *Hierarch* is a lightweight Python module for nonparametric hierarchical bootstrapping and permutation testing based on NumPy [19] and Numba [20]. In this paper,

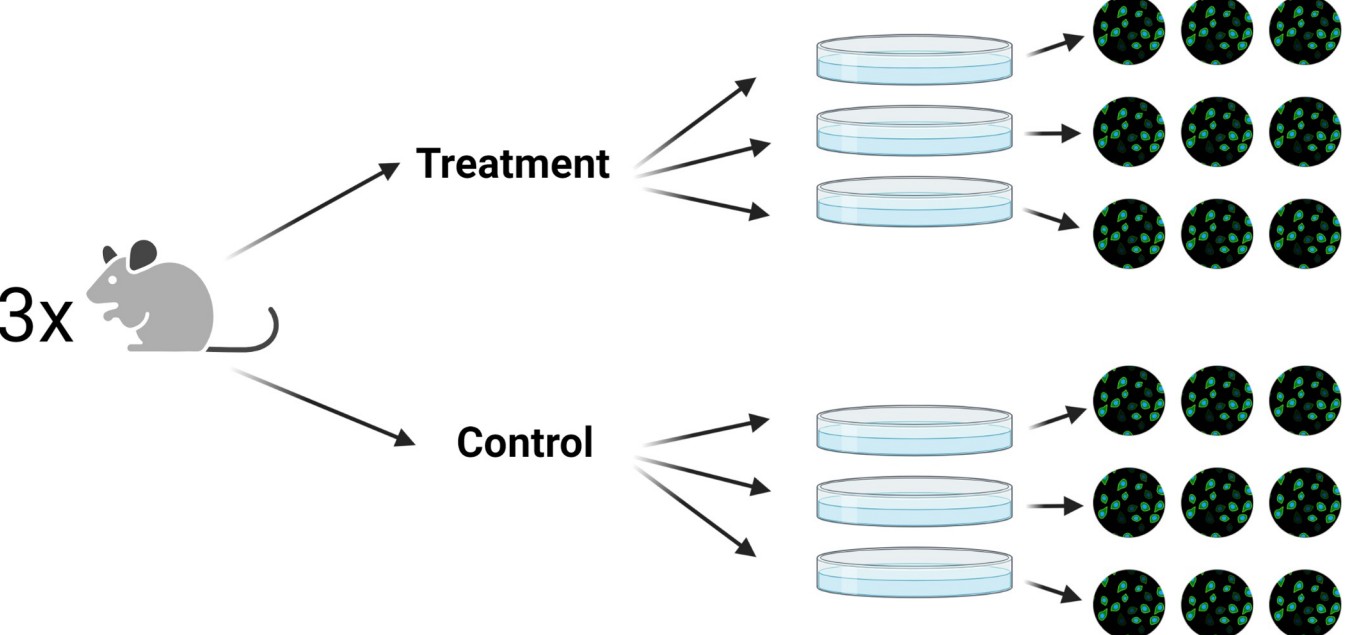

**Fig 1. Hierarchical experiments are common in biomedical research.** An example of a hierarchical experiment is an imaging experiment, where cells are isolated from donors, which are then treated in separate wells, which are then imaged under a microscope. These experimental designs are common in many fields of research, but especially so in molecular biology, imaging, and neuroscience. Figure was created using BioRender.com.

we validate the Type I and Type II error rates of hierarchical randomization tests in *Hierarch* against asymptotic tests and walk readers through their usage. We compare the properties of these tests in simulation studies with the small sample sizes typical of biological experiments (n = 3 to 4 clusters), with different underlying population distributions, and with varying levels of hierarchy. We conclude that hierarchical resampling-based hypothesis tests are powerful, maintain better control of Type I error rates than asymptotic tests in a wide variety of conditions, and enable researchers to smoothly include multiple levels of clustering beyond the classic "biological replicate" and "technical replicate" dichotomy.

## How can you tell if your data is hierarchical?

Hierarchical data arises from one (or both) of two design issues (**Fig 2**) [21]. The first issue to consider is hierarchical sampling, in which the sampled entities and the treated entities are not the same. For example, a researcher studying macrophages collects those macrophages by drawing a random sample of blood from a random sample of mice, then applying treatments to different wells in a 6-well place of the macrophages. The researcher has to account for the fact that random errors are introduced by *both* the mouse and the well–each mouse has different genetics and a different immune system, which introduces random errors to the measurement. Similarly, each well is delivered a slightly different number of cells and a slightly different amount of drug. Failure to account for both of these levels of hierarchy can result in unwarranted precision in the estimate of a treatment effect, which can fail to reproduce when the experiment is repeated in other mice.

The second design issue to consider is hierarchical assignment of treatment groups–or when the treated entities and the observed entities are not the same [21]. For example, the researcher divides each mouse's macrophages into six different wells and treats three of them with Treatment A, while the other three are treated with Treatment B. Then, the researcher

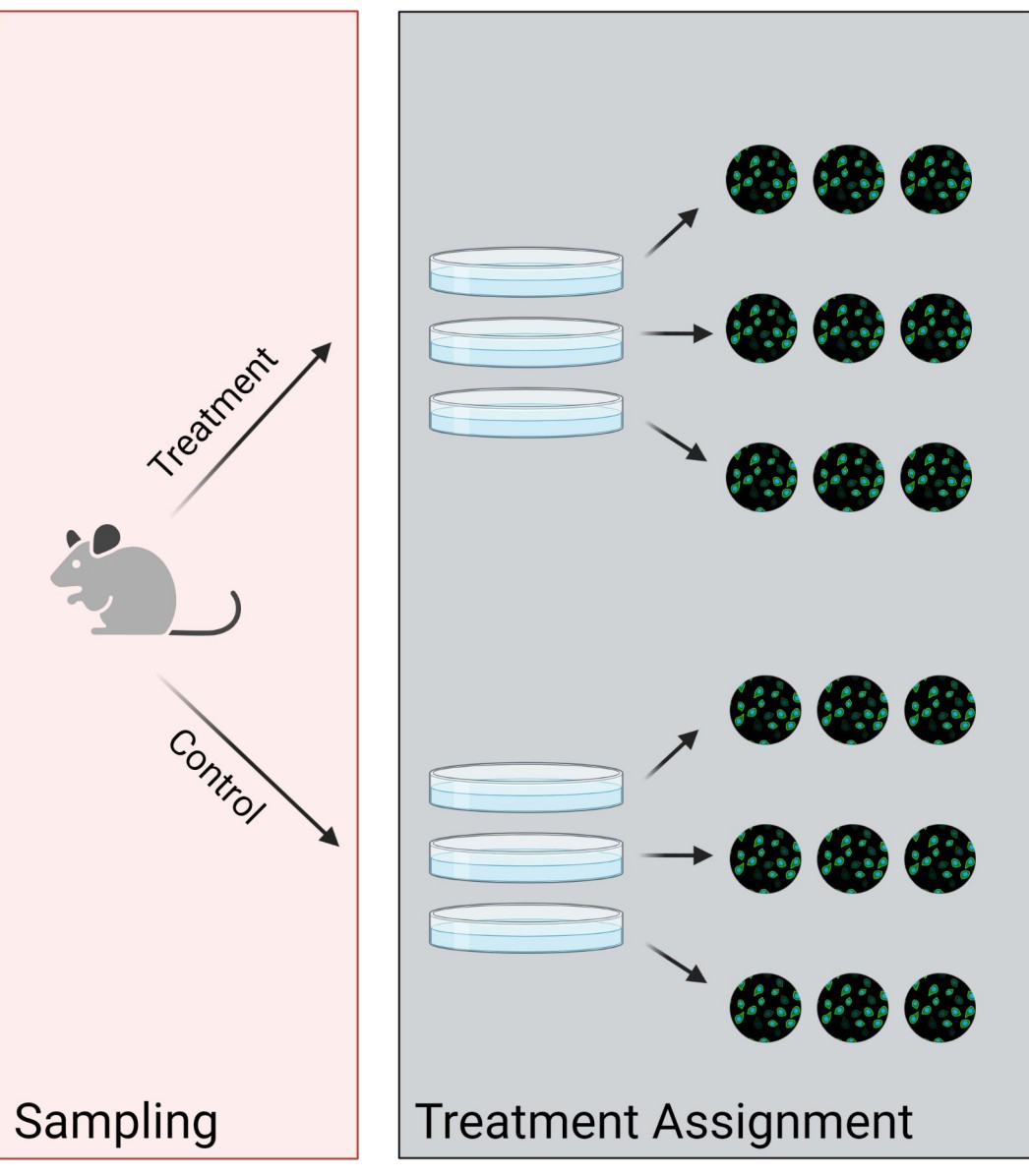

**Fig 2. Hierarchy arises during sampling and treatment assignment.** Hierarchy due to sampling occurs when the sampled entities and the treated entities are not the same. Hierarchy due to treatment occurs when the treatment entities and the observed entities are not the same. Figure was created using BioRender.com.

performs an imaging experiment in which they look at several macrophages in each well. Because two macrophages in the same well are subjected to the same random errors in environment and treatment, they are much more similar than two macrophages in different wells. Again, failure to account for these levels of hierarchy can result an overly precise estimate of the treatment effect that disappears upon replication.

Under this framework, the vast majority of molecular biology and neuroscience experiments have at least three, if not four levels of hierarchy. Unfortunately, these design issues are difficult (and sometimes impossible) to avoid due to reasons of cost, ethics, or sample availability. However, by using statistical tools that understand hierarchical data, researchers can compute robust effect sizes that do not over- or under-estimate their confidence.

## Strategy for non-parametric analysis of hierarchical data

Permutation tests are a natural way to test the null hypothesis that a treatment has no effect, and the two samples are drawn from the same distribution, or the strong null hypothesis. Rather than using a theoretical null distribution, a permutation test builds a null distribution by shuffling the treatment labels in the data and recomputing the value of the test statistic. A permutation test assumes global exchangeability–that is, each observation was randomly assigned to one treatment or the other. Importantly, the null distribution in a permutation test is *only* conditioned on the observed data and the experimental design, so no unverifiable assumptions are made about the underlying data-generating process. For this reason, design-based permutation tests have been called the "platinum standard" [17] of statistical analysis that ought to be given the "right to first refusal" [13] when choosing an analysis for a given experiment. Permutation tests are computationally intensive; however, they have become more and more practical as personal computers have gotten faster.

Permutation tests face two key challenges when performed on hierarchical data. First, hierarchical data violates the basic assumption of global exchangeability [22]. That is, while the labels of "treatment" vs. "control" are exchangeable under the null hypothesis, cells from different wells are not exchangeable. Again, this is because cells in the same well are subject to the same random errors at the well level and are expected to be more similar than cells from different wells. This problem can be avoided by only permuting on exchangeable levels (**Fig 3A**). When analyzing experimental data, this means permuting the level at which the treatment was administered. This leads us to the second problem. When there are only a small number of available permutations and the researcher wishes to perform a two-tailed test, the empirical null distribution is too coarse for the $p < 0.05$ significance level. For example, with n = 3 in each group for a two-tailed hypothesis test, the smallest false positive rate that can be achieved is 0.1. At n = 4 per treatment, the only alpha below 0.05 is 0.028. Only at n = 5 per treatment or more can the experimenter control alpha at values close to 0.05. We note that the most robust way around this issue is to perform experiments with at least n = 5 per treatment. However, it is sometimes impossible to acquire more samples, for example in cases where samples are sourced from human subjects. Ordinarily, this leaves the researcher stuck between a rock and a hard place—either they have to go with the strong assumptions of an asymptotic test (which

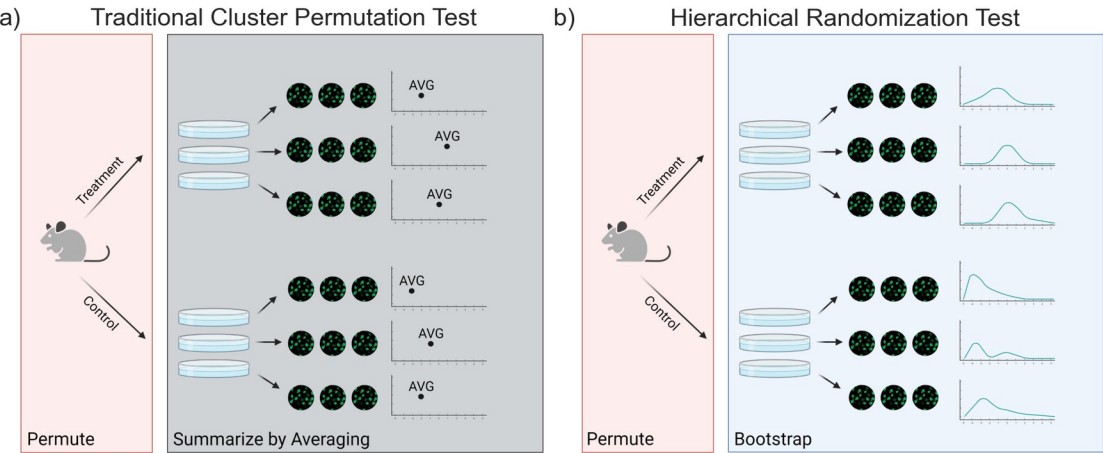

**Fig 3. Hierarchical randomization combines permutation and bootstrapping to perform a hypothesis test.** (A) By averaging, traditional cluster-based permutation tests shuffle discard information from the levels of hierarchy that arise due to treatment assignment. (B) Hierarchical randomization tests use bootstrapping to compute posterior distributions of the mean for each treated sample, thereby using all of the data collected to compute a p-value. Figure was created using BioRender.com.

are doing at least as much work as the data is) or accept that they cannot achieve $p < 0.05$ with a nonparametric test.

In this example, a traditional cluster permutation test would involve summarizing the observations in each well by taking the average, then permuting the treatment labels to form a null distribution. With only 6 total wells, however, there are only 20 possible permutations, so the minimum two-tailed *p*-value that can be computed is 2/20, or 0.1. Instead, we propose a test that shuffles posterior distributions of the cluster means rather than merely the point estimates of the cluster means (**Fig 3B**). To estimate these posterior distributions, we utilize another resampling-based method. The nonparametric bootstrap, developed by Efron [23] and extended by many others [2,24–26], is an attractive method to nonparametrically estimate the posterior distribution of each cluster mean in this situation. The bootstrap procedure involves resampling the within-cluster observations with replacement and recomputing the mean many times (say, 1000, as there are now $27^6$ possible combinations of bootstrapped wells), resulting in a distribution of means that, importantly, reflect the standard error and skew of the original observations. Then, each set of bootstrap means is shuffled some number of times (in this case, 20) and the test statistic is recomputed with every shuffle. The *p*-value is the fraction of these t statistics that are as or more extreme than the observed t statistic. This strategy, which performs bootstrap aggregation of several permutation tests [27], enables researchers to incorporate the observed within-cluster variability into the hypothesis test and control alpha at 0.05 for datasets with as few as 6 clusters.

## Methods

All hierarchical randomization tests were performed using *Hierarch* version 1.1.1.

### Data formatting

*Hierarch* uses the structure of the input data to infer the resampling strategy that should be used. As a result, users must ensure their data is formatted in this manner (Table 1). Briefly, *Hierarch* assumes that the hierarchical design of an experiment can be inferred by reading the

**Table 1. Formatted dataset corresponding to the design in Fig 2.**

| Mouse | Treatment | Well | Image | Measured Values |
|---|---|---|---|---|
| 1 | 1 | 1 | 1 | 1.35 |
| 1 | 1 | 1 | 2 | 7.84 |
| 1 | 1 | 1 | 3 | 55.2 |
| 1 | 1 | 2 | 1 | 124.4 |
| 1 | 1 | 2 | 2 | 12.2 |
| 1 | 1 | 2 | 3 | 11.1 |
| 1 | 1 | 2 | 1 | 4.444 |
| 1 | 1 | 2 | 2 | 76.3 |
| 1 | 1 | 2 | 3 | 395.3 |
| 1 | 2 | 3 | 1 | 2.1 |
| 1 | 2 | 3 | 2 | 1.199 |
| 1 | 2 | 3 | 3 | 4.4 |
| 1 | 2 | 3 | 1 | 3.3 |
| 1 | 2 | 3 | 2 | 32.2 |
| 1 | 2 | 3 | 3 | 8.8 |
| . . . | . . . | . . . | . . . | . . . |

columns left to right. For example, we will consider the hierarchical design shown in **Fig 2**. The experimental design for this experiment can be summarized as follows,

$$Data = Level_1 + Level_2 * \beta + Level_3 + Level_4$$

where $Level_1$ is Mice, $Level_2$ is the Treatment, $Level_3$ is the Wells, and $Level_4$ is the Images. Images are nested within Wells, which are nested within Treatments, which are nested within Mice. To perform a hierarchical randomization test, the data should be organized in the following table (which is fed to *Hierarch* as a Pandas DataFrame):

*Hierarch* uses the left-to-right column order of this table to infer the above experimental design. Currently, *Hierarch* assumes that the final column of the table contains the values of the dependent variable.

## Crossed factors

In fact, the Mouse and Treatment factors in **Fig 2** are crossed (each Mouse experiences each Treatment and vice versa). However, permutation tests for crossed designs can be constructed by treating the factor under study as nested within other factors on the same level [28]. This strategy can be extended to studying any number of factors on the same level. For example, consider a three-way crossed design in **Table 2** (other levels of hierarchy are omitted for clarity). Factor 3 can be analyzed with an exact permutation test by rearranging the design matrix on the left side of the table into the design matrix on the right side of the table.

## Interaction effects

Interaction effects cannot be examined with an exact permutation test, as there are no possible permutations within the main effects [28]. However, *Hierarch* can account for the presence of an interaction effect when examining a main effect. This can be done simply by adding a column after the effect of interest containing the interaction term (left side of **Table 3**). Alternatively, users can just add a duplicated column (right side of **Table 3**), as *Hierarch* constructs the appropriate by examining the number of unique rows, which are identical.

## Simulations

For Type I error rate control studies, p-values for t tests and linear regression were computed using scipy.stats. For confidence interval simulations, parametric confidence intervals were generated using the statsmodels package. We performed between 20,000 and 100,000 resamples per test depending on the total number of available permutations. The largest tests

**Table 2. For experiments with several factors on the same level (left side of table), exact permutation tests can be constructed by concatenating all but one of the factors and treating the leftover factor as nested within the others (right side of table).**

| Factor 1 | Factor 2 | Factor 3 | Factor 1 \| 2 | Factor 3 |
|---|---|---|---|---|
| 1 | 1 | 1 | 1–1 | 1 |
| 1 | 1 | 2 | 1–1 | 2 |
| 1 | 2 | 1 | 1–2 | 1 |
| 1 | 2 | 2 | 1–2 | 2 |
| 2 | 1 | 1 | 2–1 | 1 |
| 2 | 1 | 2 | 2–1 | 2 |
| 2 | 2 | 1 | 2–2 | 1 |
| 2 | 2 | 2 | 2–2 | 2 |

**Table 3. While *Hierarch* cannot produce *p*-values and confidence intervals for interaction effects, it can account for interactions when estimating main effects.**

| Factor 1 | Factor 2 | Factor 1:2 | Factor 1 | Factor 2 | Factor 2 |
|---|---|---|---|---|---|
| 1 | 1 | 1–1 | 1 | 1 | 1 |
| 1 | 2 | 1–2 | 1 | 2 | 2 |
| 2 | 1 | 2–1 | 2 | 1 | 1 |
| 2 | 2 | 2–2 | 2 | 2 | 2 |

(100,000 resamples) took less than 200 milliseconds each. In each test, we set alpha to 0.05 and simulated 10,206 datasets (729 on each of 14 cores of an Intel i9-9940X CPU) according to one of the two following data-generating models:

$$Data = Level_1 * \beta + Level_2 + Level_3$$

$$Data = Level_1 + Level_2 * \beta + Level_3 + Level_4$$

In each simulation, we generated the cluster baseline and the individual values with either normal, lognormal, Pareto, or gamma random variables. To demonstrate the general applicability of hierarchical hypothesis testing, we varied the ratio of within-cluster variance to total variance (intraclass correlation). The results of these simulations were evaluated for Type I error rate control–essentially, what percentage of the simulated datasets and hypothesis tests returned a *p*-value below 0.05 when there was no true difference between the datasets?

Confidence intervals are calculated using the test inversion procedure discussed in Manly [29]. Briefly, the bounds of the hypothesis test's rejection region are found using an iterative approach. These bounds are then unstudentized back to the units of the β coefficient. Each iteration was allowed to perform between 1,000 and 10,000 shuffles with a maximum of 10 total iterations per bound. These simulations were repeated for varying values of β. The results of these simulations were evaluated for coverage probability–essentially, what percentage of the 95% confidence intervals contain the true value of β?

## Description of the hierarchical randomization test

To explain our algorithm, we consider again the above dataset consisting of two **treatments**, three **wells** each, and three **images** each (**Fig 2**, simulated data in **Fig 4A**). The researcher seeks

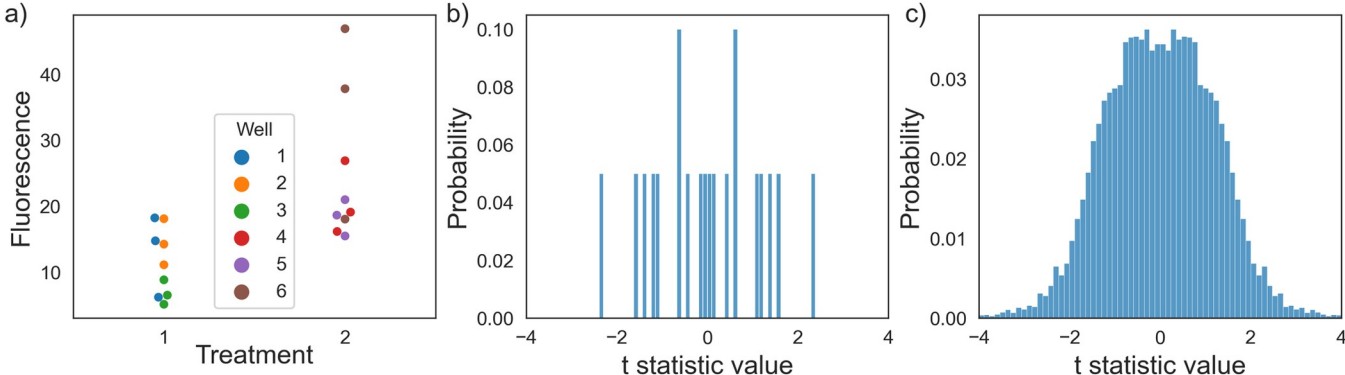

**Fig 4. Hierarchical randomization uses bootstrapping to construct the empirical null distribution.** (A) Simulated data corresponding to an experiment in **Fig 2**. (B) A traditional cluster permutation test would only have twenty possible permutations, resulting in the smallest calculable two-tailed p-value being 0.1. (C) Hierarchical resampling constructs a full empirical null distribution, resulting in a p-value of 0.0429.

to test the null hypothesis that the treatment had no effect on the mean fluorescence intensity of each image. The hypothesis test proceeds as follows:

1. Identify the treated level (in this case, **wells**).

2. Aggregate the data (by averaging) so that each treated entity corresponds to a single $y$-value. In this case, calculate the **per-well** mean fluorescence.

3. Compute the value of the test statistic of interest. This is the observed test statistic.

4. From the original data, resample the treated level via bootstrapping. In this case, generate new **wells** by resampling each one from its **images**.

5. Repeat step 2.

6. Permute the **treatment** labels and recompute the test statistic. Add this value to the null distribution.

7. Repeat step 6 a large number of times (default 1000). This experiment only has 20 possible permutations, so we would specify permutations = "all" in the hypothesis_test function to enumerate all of them. At this point, the null distribution is shown in **Fig 4B**.

8. Repeat steps 2 through 7 a large number of times (default 100). If there are few possible permutations, the number of bootstraps can be increased to improve the stability of the computed $p$-value. For this experiment, we performed 500 bootstraps. At this point, the null distribution is shown in **Fig 4C**, as we have computed 20 x 500 = 10,000 resampled test statistics.

9. Determine what fraction of the null distribution is as or more extreme than the observed test statistic. This number is the two-tailed $p$-value.

A traditional cluster permutation test would only be able to produce a null distribution containing at most 20 possible values (**Fig 4B**), but hierarchical randomization generates a full null distribution without making distributional assumptions about the data (**Fig 4C**). More generally, the algorithm deals with each level of hierarchy in one of two ways. For hierarchy arising due to treatment assignment, the algorithm uses nonparametric bootstrapping to estimate the sampling distribution of the mean. For hierarchy due to sampling, the algorithm restricts the number of possible permutations such that only "within-cluster" permutations are possible. This procedure mimics the data-generating process under the null hypothesis (that is, the hypothesis that the treatment did nothing at all). First, each well is resampled from its fields of view and then randomly assigned to one of the two "treatment" labels. Using *Hierarch*, this procedure is fully automatic—once the researcher has specified their experimental design by organizing their data, the algorithm will produce a $p$-value without requiring any further input. Moreover, the algorithm infers the correct resampling plan for any hierarchical experimental design. If a researcher pre-commits to using hierarchical randomization as their analysis tool of choice, they have eliminated an important researcher degree of freedom—choice of hypothesis test—whilst retaining the flexibility to analyze a wide range of experimental designs.

## Caveats

The most important assumption this test makes is that the labels being shuffled are exchangeable under the null hypothesis. In other words, it assumes that the clusters attached to the labels were assigned randomly. The second, weaker assumption is that the observations have

similar distributions (though not necessarily normal). This is a weak assumption because by using an approximately pivotal test statistic (such as the t statistic) [16,30], the assumption of homogeneity of variances does not have to be fulfilled for this test to maintain control of Type I error rate. However, with very few clusters, this test can be sensitive to heterogeneity of variances (see simulation study below).

An important consideration with this approach is that bootstrapping is only appropriate when the within-cluster data points represent a random sampling of possible within-cluster values. For example, an imaging experiment might involve taking images of several fields of view within a well and measuring some per-cell quantity in each image. In this case, the fields of view are randomly sampled from all fields of view in the well (as there are fields of view that were ignored) and therefore can be resampled, but cells within each field of view are not randomly sampled (as every cell in a given field of view is measured) and therefore should not be resampled via bootstrapping. For a deeper discussion of this, see van der Leeden, et al. [9].

Another consideration to using these resampling techniques is that the permutation test is no longer exact—there are usually a much larger number of possible resamples than can reasonably be calculated (though in this simple case, it is possible). However, by performing a large number of resamples and using an appropriate test statistic, this approximate test will have size close to 0.05 and good power while not requiring the researcher to make distributional assumptions about their dataset. To demonstrate the flexibility of hierarchical randomization tests, we will discuss the analysis of three datasets.

## Construction of a studentized covariance test statistic

When constructing a randomization test for some parameter, the test is only guaranteed to be exact for the null hypothesis of the distributions being equal. To maintain a Type I error rate of 5% for the more general null hypothesis that the parameter is equal, the test statistic must be at least approximately pivotal–that is, its distribution does not depend on unknown parameters, such as the population standard deviation. Approximately pivotal statistics can be constructed following the procedure of Janssen [16], which was expanded by Chung and Romano and others [30–33]. This is done by dividing the comparison of interest by an estimate of its standard error. As an illustrative example, we will discuss the construction of the t statistic. We consider a linear equation describing a data-generating process.

$$Treatment * \beta + Cluster + Individual = Data. \tag{1}$$

When there are only two treatment groups, β can be estimated using Eq 2,

$$\beta = \bar{B} - \bar{A} \tag{2}$$

$\bar{A}, \bar{B}$ are the means of group A and B, respectively.

Student's t test is a hypothesis test against the null hypothesis that β = 0. The t test is based on the t statistic, which is given by the following equation,

$$t = \frac{\bar{B} - \bar{A}}{\sqrt{\frac{s_B^2}{n_B} + \frac{s_A^2}{n_A}}} \tag{3}$$

- $\bar{A}, \bar{B}$ are the means of group A and B, respectively.

- $s_A^2, s_B^2$ are the sample variances of group A and B, respectively.

- $n_A, n_B$ are the number of samples in group A and group B.

This is equivalent to the following expression:

$$t = \frac{\beta}{s.e.(\beta)}$$

- $\beta$ is the estimator for the slope in Eq 5.

- $s.e.\ (\beta)$ is the standard error of the estimator $\beta$.

This is a general approach for constructing an asymptotically normally distributed test statistic (or a Wald-like statistic). Because of this property, when a Wald-like statistic is used as the test statistic in a randomization test, the test gains asymptotic validity against unequal variances between treatment conditions and gives the researcher the ability to make directional conclusions. However, the t statistic can only be used as a test of $\beta = 0$ when there are only two samples. Instead, we can express $\beta$ as a ratio between the covariance of X and Y and the variance of X:

$$\beta = \frac{Cov(X, Y)}{Var(X)}$$

- $X$, $Y$ are the treatment condition and observed data, respectively.

In a randomization test, we are merely shuffling the relationship between X and Y. Therefore, the variance of X is constant during the shuffling procedure. We can therefore construct a Wald-like test statistic for $\beta$ using the covariance of X and Y, which is based on the work of DiCiccio and Romano [34]:

$$T = \frac{Q}{\sqrt{S^2(Q)}} \tag{4}$$

- $Q$ is the sample covariance of $X$ and $Y$.

- $\sqrt{S^2(Q)}$ is standard error of the sample covariance of X and Y, or the square root of the sample variance of the sample covariance of X and Y.

The sample covariance of $X$ and $Y$, $Q$, is given by Eq 5:

$$Q = \frac{n\mu_{1,1}}{n - 1} \tag{5}$$

- $n$ is the number of total observations.

- $\mu_{1,1}$ is the population covariance of $X$ and $Y$, otherwise known as the first product central moment of $X$ and $Y$. This is computed with Eq 6:

$$\mu_{r,t} = \frac{1}{n}\sum_{i=0}^{n}(X_i - \bar{X})^r(Y_i - \bar{Y})^t \tag{6}$$

To compute the sample variance of $Q$, it is helpful to start with the population variance. For any distribution with defined moments, the population variance of the sample covariance is

expressed by Eq 7:

$$\sigma^2(Q) = -\frac{(-2+n)\mu_{1,1}^2}{(-1+n)n} + \frac{\mu_{0,2}\mu_{2,0}}{(-1+n)n} + \frac{\mu_{2,2}}{n} \tag{7}$$

- $n$ is the number of total observations.

- $\mu_{r,t}$ represent product central moments of $X$ and $Y$ given by the Eq 6.

Eq 7 represents a biased estimator for the variance of $Q$, however. The unbiased estimator for the variance of Q, or the sample variance of Q, is prone to numerical instability (**S1 Text**), so instead, we use the following bias-corrected approximation for the sample variance of Q,

$$S^2(Q) = \frac{1}{n - \frac{3}{2}}\left(-\frac{n^2(n-2)\mu_{1,1}^2}{(n-1)\left(n-\frac{7}{4}\right)^2} + \frac{n^2\mu_{0,2}\mu_{2,0}}{(n-1)^3} + \frac{n\mu_{2,2}}{n-\sqrt{2}}\right) \tag{8}$$

- $n$ is the number of total observations.

- $\mu_{r,t}$ represent product central moments of $X$ and $Y$ given by the Eq 6.

Using Eq 8 as an estimator for the variance of the sample covariance of X and Y, we can use the Wald-like statistic in Eq 4 as the basis of a hierarchical randomization test. In the simulation study below, we will investigate the properties of this test against nonnormality and heteroscedasticity.

## Results

### Example 1: Three-level mouse socialization experiment

First, we consider the behavioral assay shown in **Fig 5A**. Here, the researcher has a control group and treatment group of four mice each. Each mouse performs 500 trials of a behavioral assay to test the hypothesis that the treatment causes an increase in socialization duration. The multilevel design (treatment -> mouse -> trial) of this experiment lends itself to a hierarchical hypothesis test. Furthermore, given that the units of the measurement (seconds) are bounded by zero, we have good motivation to try a test that does not assume normality. We consider the following data-generating model:

$$Level_1 * \beta + Level_2 + Level_3 = Socialization\ Duration \tag{9}$$

where $Level_1$ is the treatment condition, $Level_2$ is mice, and $Level_3$ is trials.

The researcher seeks to estimate the treatment effect $\beta$ and calculate a p-value against the null hypothesis that the treatment effect is 0. Each mouse, however, has an individual random constant that reflects mouse-to-mouse variation in baseline socialization duration. While the terms in the model can be written in any order, it can be helpful to structure the equation in the same order as the actual hierarchical experiment. In this case, treatments are assigned to mice, which are measured 500 times each. Simulated data for this experiment is shown in **Fig 5B** and **5C** with a true effect size of 1 second. The researcher can organize their data into the "long" format presented in **Fig 5B** (visualized in **Fig 5C**), where each column corresponds to one of the terms in the model (treatment, mouse, trial number). Given that the input data is organized such that the column order mimics the experimental design, *Hierarch's* hypothesis_test function will conduct the appropriate resampling plan by default.

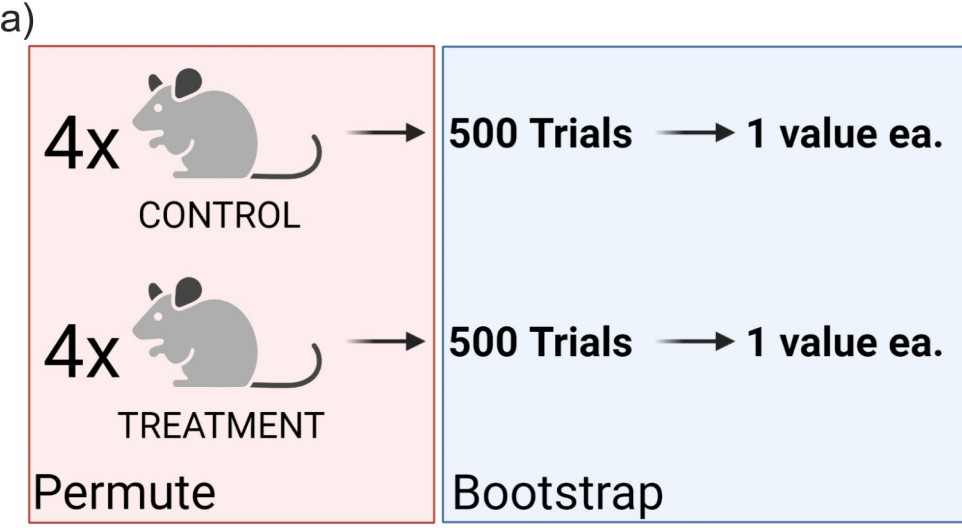

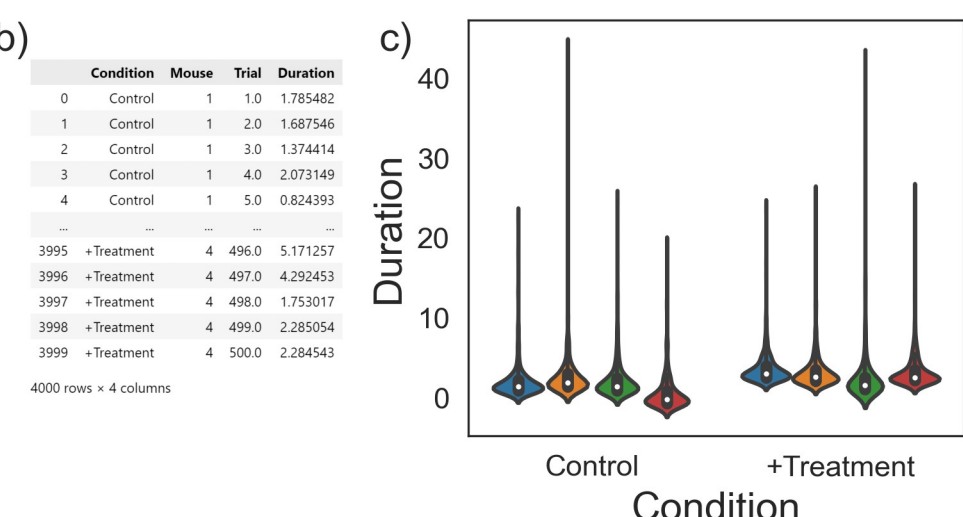

**Fig 5. Using hierarchical randomization to analyze a mouse behavioral study.** (A) Simulated data corresponding to an experiment with two treatments, four mice each, 500 trials each. (B) A table of raw data collected in this study. By organizing the input data into columns corresponding to the experimental design, *Hierarch* can automatically infer the correct resampling plan for the dataset. (C) Violin plots illustrating the skewed nature of the dataset. p = 0.037, hierarchical randomization, p = 0.038, Student's t test. Panel (A) was created using BioRender.com.

According to the hierarchical randomization algorithm, treatment labels are permuted only at the "mouse" level—individual behavioral trials are never exchanged between different mice. This ensures that the test does not break the dependence structure that exists in the dataset. Instead, uncertainty in the mouse-level mean for the behavioral trials is represented via bootstrapping. *Hierarch* performs 35,000 resamples (500 bootstraps with 70 permutations each) in less than 200 milliseconds and generates a two-tailed *p*-value of 0.037, indicating a statistically significant difference. In this case, we note that the hierarchical randomization *p*-value is similar to the *p*-value calculated by a two-sample t-test after averaging the 500 trials for each mouse ($p = 0.038$). Given the large number of trials, the standard error of the mean for each mouse is quite low, so computing an average socialization duration for each mouse does not throw

away much information. Rather, we use this as an illustrative example to show that even when another test might be appropriate, hierarchical randomization produces a similar *p*-value, but can be applied to more complicated datasets without much trouble.

While *p*-values are useful, they are best paired with a measure of effect size [35,36]. Generally speaking, accompanying the effect size with a confidence interval gives readers more information with which to interpret experimental results. One trouble with computing confidence intervals for effect sizes is that few methods actually maintain the nominal coverage when sample sizes are small– 95% confidence intervals often do not contain the true value exactly 95% of the time [37–39]. Hierarchical randomization tests can be inverted to form confidence intervals that are very close to exact. This is a key advantage over the t-test for non-normal data–t-intervals, which are quite robust for small samples, tend to be conservative for non-normal data and will produce too-wide confidence intervals. As we will show in the simulation study below, the 95% confidence intervals produced by *Hierarch's* confidence_interval function do indeed contain the true effect size 95% of the time, even for datasets with as few samples as this one. In this case, *Hierarch's* 95% confidence interval on the effect size is (0.179, 2.697) while the corresponding t-interval is (0.0166, 2.607). This is an interval around the beta coefficient in Eq 1, which we simulated with a value of 1.

## Example 2: Four-level imaging experiment

Next, we consider the motivating example from above: a paired experimental design common in molecular biology and neuroscience. Here, the researcher is interested in testing the effects of a drug on neuronal firing rate in a culture model. From each of three pregnant mice, the researcher prepares six separate neuronal cultures in a six-well plate. In each plate, the researcher treats three wells with the drug of interest and gives three a vehicle control. Then, the researcher performs a current-clamp experiment to measure the firing rate of three neurons in each well (**Fig 6A and 6B**). We consider a data-generating model as follows:

$$Level_1 + Level_2 * \beta + Level_3 + Level_4 = Firing\ Rate \qquad (10)$$

where $Level_1$ is mice, $Level_2$ is the treatment condition, $Level_3$ is wells, and $Level_4$ is images.

The researcher seeks to estimate the fixed treatment effect $\beta$ and calculate a p-value against the null hypothesis that $\beta$ is 0. We simulated the data in **Fig 6B** with an effect size of 11. Unlike the previous experiment, there is an additional constant term—we assume that not only does each well have a random baseline, but each mouse *also* has a random baseline. Despite how common this experimental design is, it is not immediately clear how best to calculate a p-value with a traditional approach. Should the researcher perform a Student's t-test with n = 9 wells in each treatment group? If each mouse has a different baseline firing rate, however, the between-mouse variance would erode the power of the test. Furthermore, the t test assumes that the treatment effect is fixed and neglects the fact that, at least on one level, the data is paired. On the other hand, aggregating the firing rates up to the treatment level and performing a paired t test with n = 3 also has little power by virtue of reducing the sample size to 3.

Two traditional options can be used in this situation: either treating each mouse as a separate experiment and combining the data in a manner analogous to an individual participant data meta-analysis or fitting a mixed effect model [40–42]. Both of these approaches require researchers to make distributional assumptions about their datasets, however. Hierarchical randomization provides a natural way to test a single hypothesis and generate a single p-value on the combined experiments—bootstrap the mean firing rate for each well from its neurons, then permute the treatment labels on the wells *within* mice [43]. In this example, there are many more possible permutations (20^3 = 8,000), so the researcher can choose to run a subset

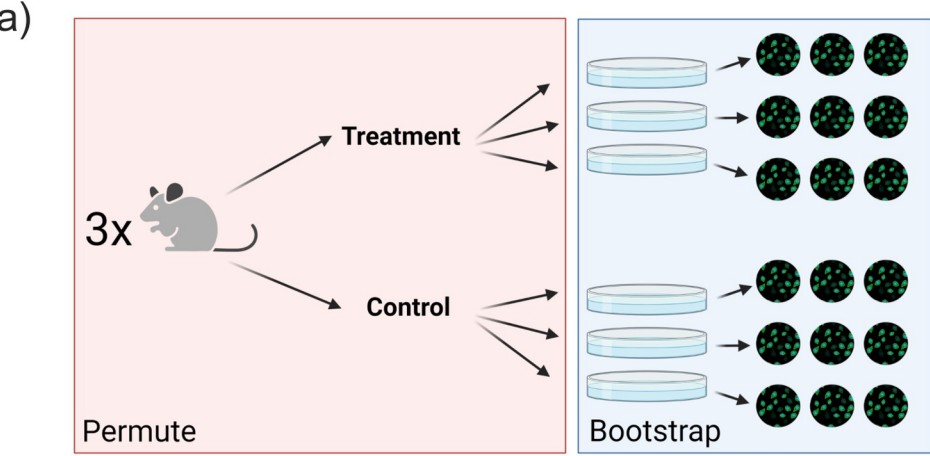

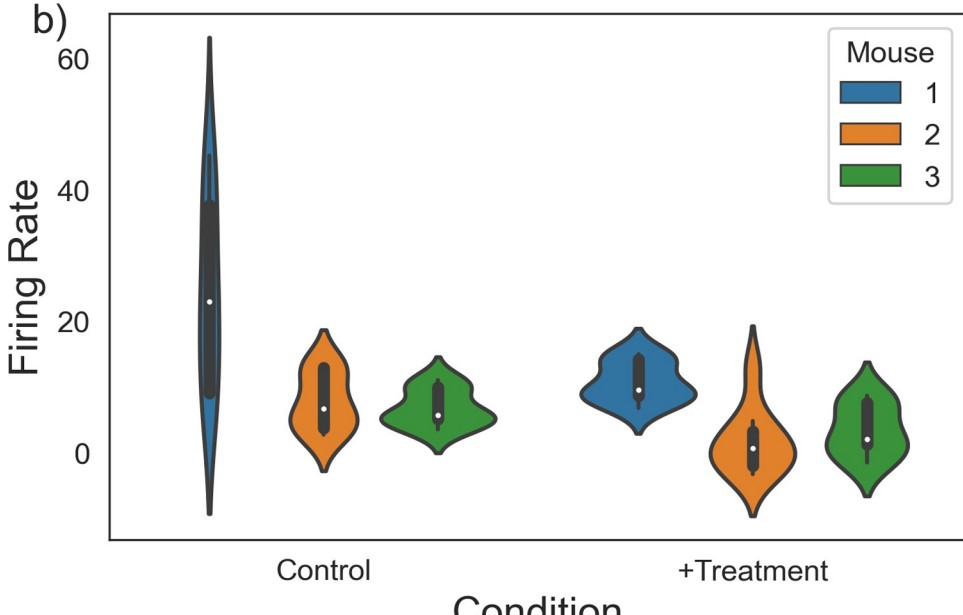

**Fig 6. Analyzing a four-level imaging experiment.** (A) An experiment with three mice, two treatments each, three wells each, and three neurons each. (B) Violin plots visualizing the dataset. p = 0.027, hierarchical randomization, p = 0.101, pooled Student's t test. Panel (A) was created using BioRender.com.

of them (100 bootstraps, 4,000 permutations each). This results in a *p*-value of 0.027 and a 95% effect size confidence interval of (1.157, 14.916), which contains the true value of 11. Pooling all of the data and performing a t test gives a *p*-value of 0.101 and a 95% confidence interval of (-2.009, 17.305). By accounting for sampling hierarchy, hierarchical randomization can be more powerful than other non-meta-analytic approaches.

Given that we have assumed a fixed treatment effect, the experiments (mice) are automatically weighted by their sample size. However, hierarchical randomization makes it simple to analyze data from a random treatment effect perspective, as well. If the researcher suspects there may be significant heterogeneity in treatment effects—perhaps the donor mice are heavily outbred, or the donors are humans–they can incorporate this heterogeneity as an

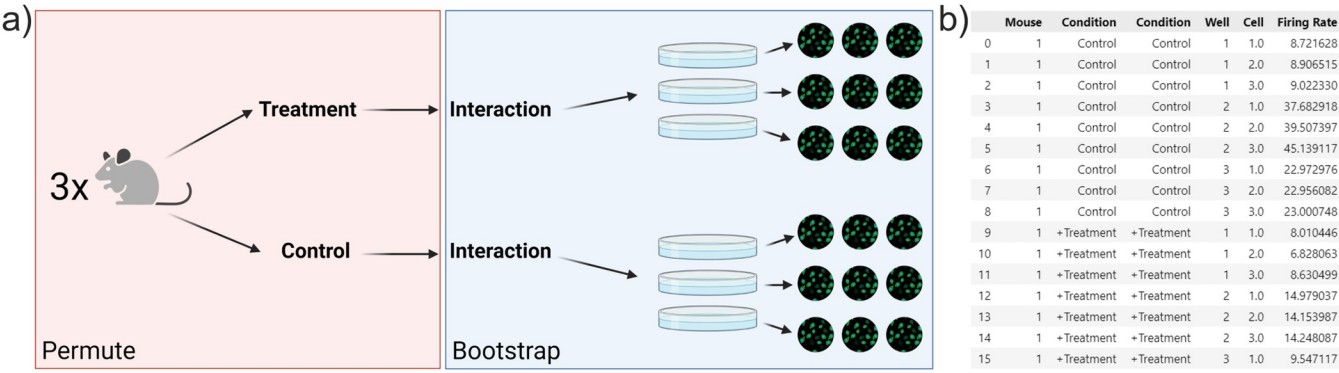

**Fig 7. Incorporating heterogeneous treatment effects.** (A) Adding a mouse-treatment interaction term to the experiment design describes heterogeneity of treatment effects. (B) By duplicating the "Condition" column in the input data, *Hierarch* will account for this interaction. Panel (A) was created using BioRender.com.

interaction effect (**Fig 7A**) [44]. The equation can be written as follows:

$$Level_1 + Level_2 * (\beta + Level_1) + Level_3 + Level_4 = Firing\ Rate$$

where $Level_1$ is mice, $Level_2$ is the treatment condition, $Level_3$ is wells, and $Level_4$ is images.

The $Level_1$ term in the interaction is not the same value as the $Level_1$ baseline constant–rather, it represents the fact that in this model, we are allowing each mouse to have a unique slope for the treatment effect. Distributing this equation gives:

$$Level_1 + Level_2 * \beta + Level_1 * Level_2 + Level_3 + Level_4 = Firing\ Rate. \tag{11}$$

This equation splits the treatment effect into two terms–an average treatment effect (Treatment x β) and a random interaction effect (Treatment x mouse). Updating the raw data to include this additional term is all that is necessary for *Hierarch* to carry out the appropriate resampling plan for the random treatment effect model. This is most easily done by simply duplicating the "treatment" column in the raw data (**Fig 7B**), which communicates to *Hierarch* that an interaction term is present.

Accounting for treatment effect heterogeneity reduces the possible number of permutations (from $20^3 = 4000$ to $2^3 = 8$). This increases the *p*-value (0.118) widens the confidence interval (0.478, 23.45, 90% confidence interval). We note that because of the severely restricted number of permutations, it is not possible to compute a 95% confidence interval for the main effect–the smallest tail probability that *Hierarch* can reliably describe is inversely proportional to twice the number of possible permutations. This makes sense in the context of a random-effect model—in order to make a precise estimate of effect size, both the "within-mouse" sample size and the number of mice must be large. Performing a large number of samples within a single mouse may yield a very precise estimate of the effect size in that mouse, but if the effect varies mouse to mouse, the overall average effect can only be accurately estimated by studying several mice. Furthermore, when assuming random treatment effects, reporting a confidence interval on the effect size is more important than ever because the average treatment effect is entirely dependent on the mix of donors. Given that, summarizing the effect size with a single point estimate is too reductive–after all, there is no single number that can describe the true effect of the drug.

### Example 3: Four-level rat behavioral study with multiple time points

Finally, we consider an experimental design with several treatments that seeks to test a single hypothesis. A researcher is interested in measuring changes in a neural population over the course of learning a task. The researcher has four rats, who are each measured on four days in the learning process. On each day, the rat attempts the task 500 times, during which some population of neurons is recorded via electrodes implanted in the rat's skull. From each recording, the researcher computes some neural population-level metric (**Fig 8A** and **8B**). The researcher considers the following model:

$$Level_1 + Level_2 * \beta + Level_1 * Level_2 + Level_3 = Firing\ Rate \tag{12}$$

where $Level_1$ is mice, $Level_2$ is the day, and $Level_3$ is trials.

We include an interaction effect to account for day-to-day variation in the population effect due to electrode drift and other changes in the mouse that are unrelated to the task at hand. As above, we want to perform a hypothesis test against the null hypothesis that $\beta = 0$.

a)

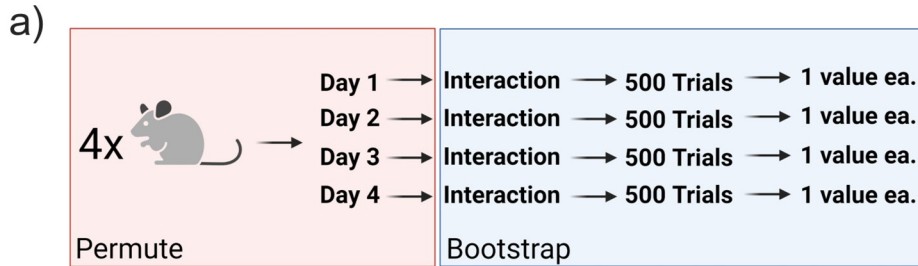

b)

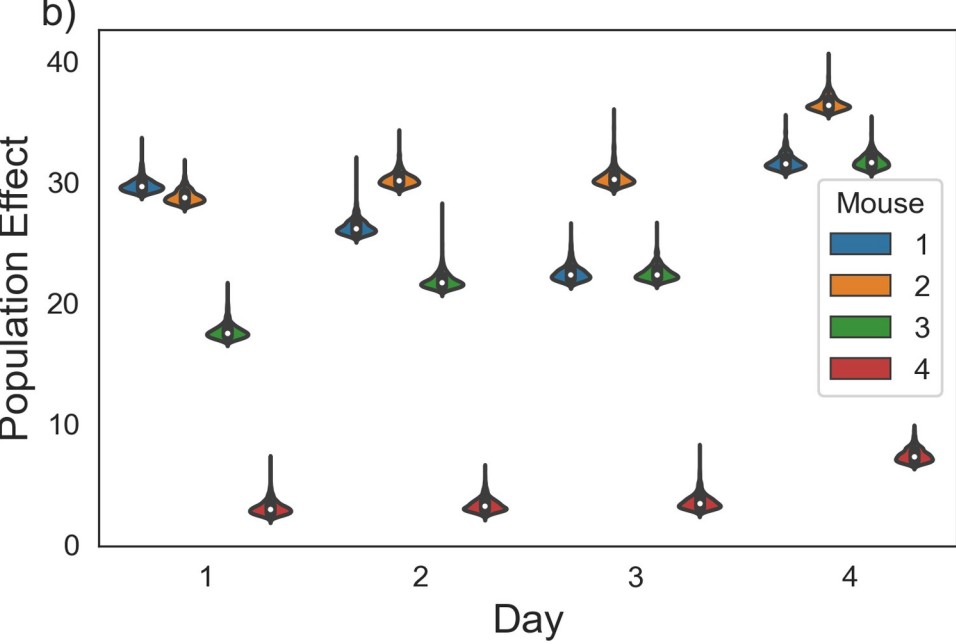

**Fig 8. Analyzing an experiment with multiple treatment conditions testing a single hypothesis.** (A) An experiment with four mice, two treatments each, three wells each, and three neurons each. (B) Violin plots visualizing the dataset. $p = 0.024$ (hierarchical randomization) for the hypothesis that there is a day-to-day increase in the population effect. Panel (A) was created using BioRender.com.

The experimental design poses another challenge, however—there are four different days. One approach could be to perform several two-sample tests between each day and the next day. However, this approach only considers a subset of the dataset at a time, and as a result loses a lot of power–none of the day-to-day comparisons are significant. Upon closer examination, this logic behind this approach is unclear–if we have a single hypothesis (that $\beta \neq 0$), why perform multiple hypothesis tests? Another option is fitting a simple linear regression or a mixed model–but we have no reason to think the errors of the neural population measure are normally distributed and, as usual, we do not have many clusters. This example motivates the construction of a studentized covariance test statistic that can be used to perform a single hypothesis test against the null hypothesis that $\beta = 0$ when there are multiple treatment groups with a hypothesized linear relationship.

This test statistic can be calculated on every shuffled dataset in a hierarchical randomization test, which provides a test against the null hypothesis that the slope for a given regressor in a linear model is equal to zero. For two-sample datasets, this test statistic has a linear relationship with the t-statistic and therefore will calculate the same *p*-value, which is demonstrated in the simulations below. In this instance, hierarchical randomization computes a *p*-value of 0.0236 and a 95% confidence interval of (0.42, 3.622), which contains the true, simulated value of $\beta = 2$.

## Simulation results

In this section, we demonstrate that hierarchical randomization successfully controls Type I error rates without being sensitive to the underlying distribution of the dataset. We were particularly interested in small studies typical in biomedical research, so we chose to consider experiments structured similarly to the case studies detailed above. For larger datasets, there are numerous other simulation studies in the literature demonstrating the good properties of randomization tests [13,30,33].

First, we examined both types of hierarchical randomization tests (using the t statistic and using the studentized covariance statistic described in Eq 5). We simulated datasets with two treatment groups in which the effect size ($\beta$) was set to zero (**Fig 9**). For each set of simulations, the fraction of hypothesis tests that returned a significant result was plotted on the *y*-axis. The shaded region represents the 95% confidence interval around a 5% Type I error rate, which each test should ideally remain in. As expected, the Student's t test is only able to robustly maintain a 5% Type I error rate when the underlying data is normally distributed, while Welch's t test is conservative in all cases. We observed that when there are 8 total clusters, the hierarchical permutation test allows good control of Type I error rate without regard for the underlying distribution. Even for very asymmetric distributions (such as the power law distribution) type I error rate could be acceptably controlled via hierarchical randomization. For 6 clusters, the hierarchical permutation test performed similarly well, but over-rejects the null for certain distributions when the between-cluster variance is much greater than the within-cluster variance. Using studentized covariance as the test statistic gave similar results (**S1 Fig**), which is expected as both statistics test for a difference in means.

We noted that the parametric tests performed quite well for experimental designs with only one level of nesting. However, when two nested levels were introduced, the parametric tests became quite conservative (**Fig 9**). This behavior necessarily results in a loss of power, making these tests inappropriate for these experimental designs. By comparison, the hierarchical randomization test suffered no similar loss of error rate control in these conditions.

Next, we investigated Type I error rate in identical conditions as above, but with unequal variances between treatment groups (1:1.5). Again, we found that the hierarchical

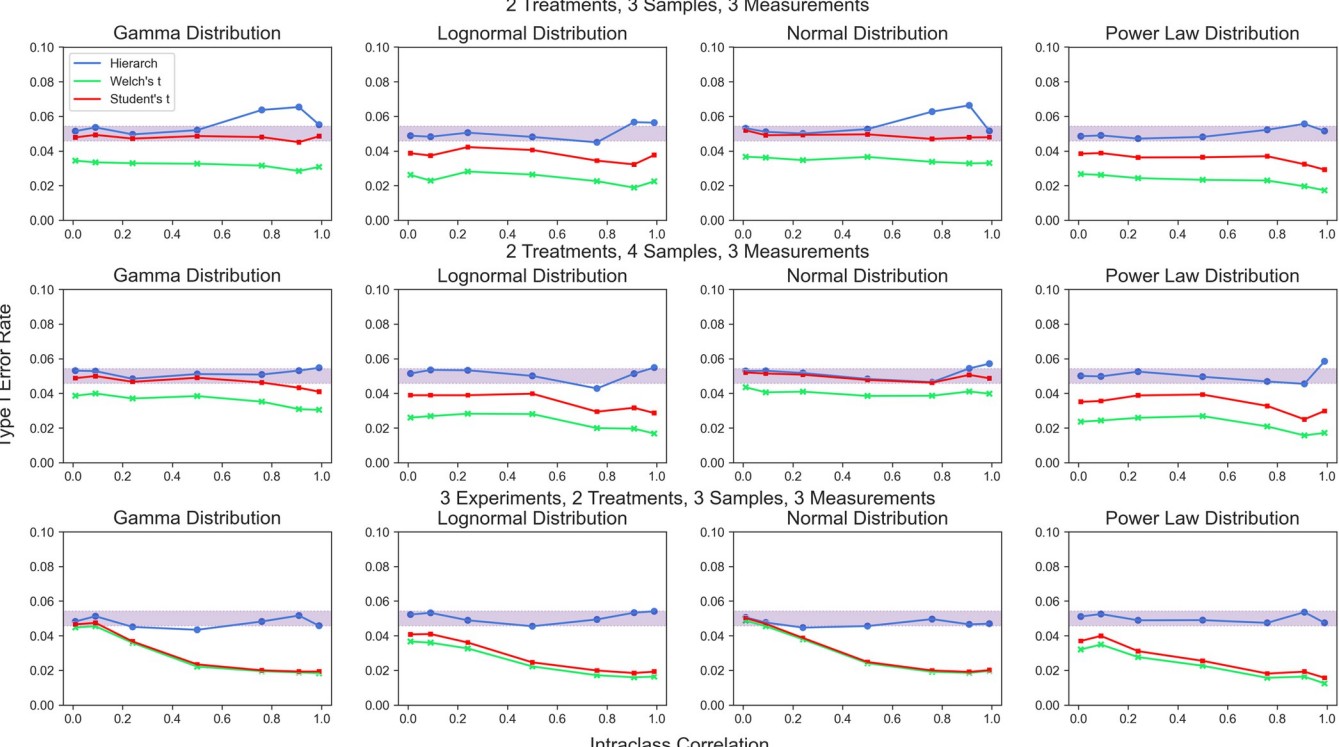

**Fig 9. Type I error rate for hierarchical randomization test based on the t statistic compared to Student's t test and Welch's t test.** Both treatment groups have equal variance. Shaded area represents the 95% binomial confidence interval around a 5% type I error rate.

randomization test afforded better control of Type I error rates than either t-test in most cases, though the performance is slightly worse for very few clusters (**Fig 10**). The usage of a pivotal test statistic gives hierarchical randomization only asymptotic validity against the weak null hypothesis [30]—as the number of treated samples increases, the better the test performs when the variances of the two samples are different. Again, the performance of the hierarchical randomization test based on studentized covariance was essentially identical to the performance of the hierarchical randomization test based on the t statistic in these conditions (**S2 Fig**).

To validate the performance of the hierarchical randomization test with multiple treatment conditions, we simulated data under identical conditions as above, but with 3 or 4 treatment groups (**Fig 11**). We compared the performance of hierarchical randomization with that of performing a Wald test on estimates from two-stage least squares (2SLS) regression, another common method for unbiased analysis of hierarchical models. We found that unlike the standard Wald test, hierarchical randomization maintains Type I error rate control when presented with non-normal errors and maintains good control in the presence of heteroscedasticity (**S3 Fig**).

Next, we examined the confidence intervals generated by hierarchical randomization as compared to those generated by parametric approaches (t-intervals for two samples, normal approximation on 2SLS for many samples). Again, we observed that the parametric confidence intervals had good coverage in experiments with a single nested level but demonstrated severe over-coverage for more complex experimental designs. We simulated data with a range of mean differences to monitor the coverage probability of 95% confidence intervals computed by hierarchical randomization (**Fig 12**). We found that inverting a hierarchical randomization test produces 95% confidence intervals that maintain nominal coverage without regard for the

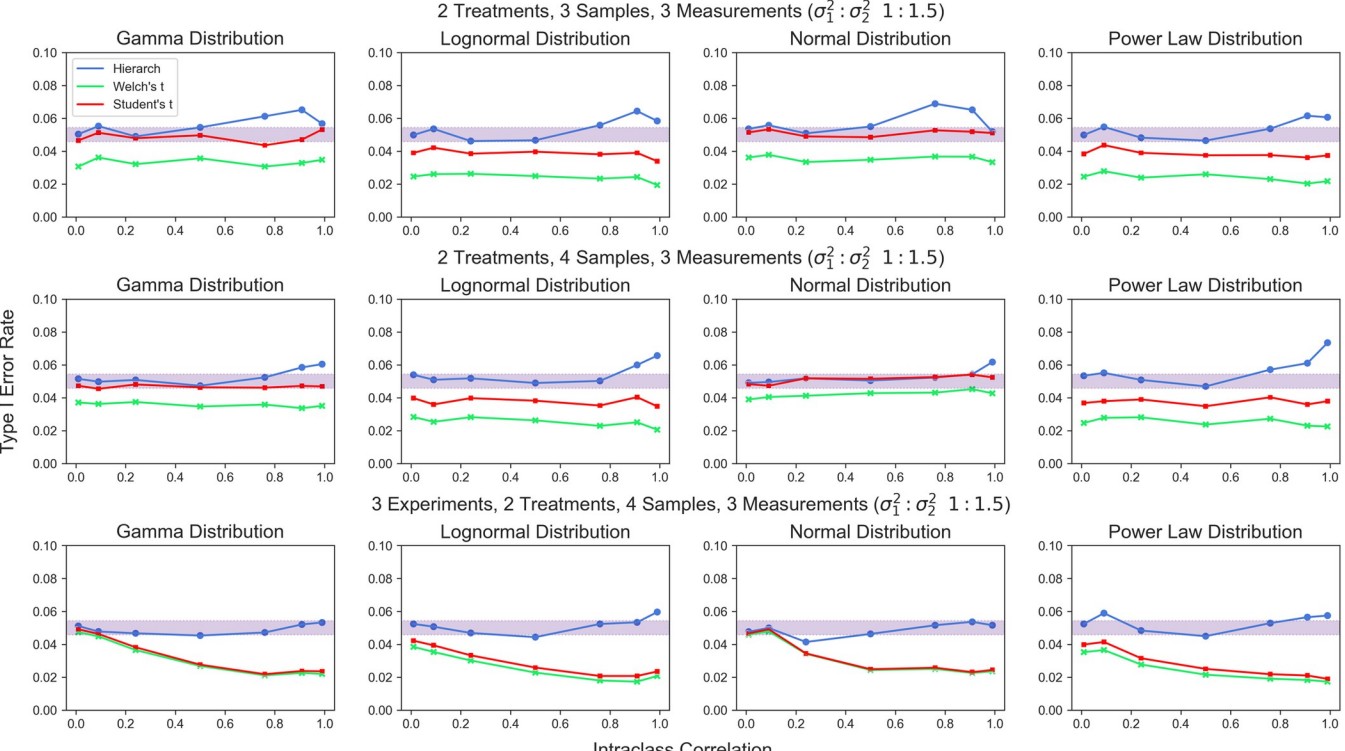

**Fig 10. Type I error rate for hierarchical randomization test based on the t statistic, Welch's t test, and Student's t test given unequal between-cluster variance.** Shaded area represents the 95% binomial confidence interval around a 5% type I error rate.

underlying distribution. The studentized covariance test statistic performs similarly well for datasets with multiple treatment groups (**Fig 13**).

## Comparison with linear mixed models

Finally, we compared the performance of *Hierarch* with that of a linear mixed model. We note that when random effects are included in a linear mixed model (LMM), unbiased estimation is no longer possible due to partial pooling, while a linear mixed model with only fixed effects is no different from 2SLS. Indeed, the incorporation of random effects is typically done when unbiased estimates are uninteresting (for example, when making predictions or aiming to regularize estimates of many coefficients). Still, linear mixed models with random effects are often used when unbiased estimation *is* desirable, so we simulated the experimental design from **Fig 2** with a variety of underlying distributions (**Fig 14**). For the mixed model, the data was fit with the following Wilkinson formula:

$$y \sim Level_1 + (1|Level_2)$$

and the *p*-values for the estimate of the $Level_1$ $\beta$ coefficients were compared to those calculated by *Hierarch*.

We were unsurprised to find that the mixed model failed to control Type I error rate. By incorporating a $Level_2$ random effect, the estimates for the $Level_2$ means are shrunk toward the global mean, which reduces their variance and thereby decreases the *p*-value of the $Level_1$ coefficient. Again, it is important to note that LMMs with random effects are not trying to make unbiased estimates, so performing hypothesis tests on these estimates

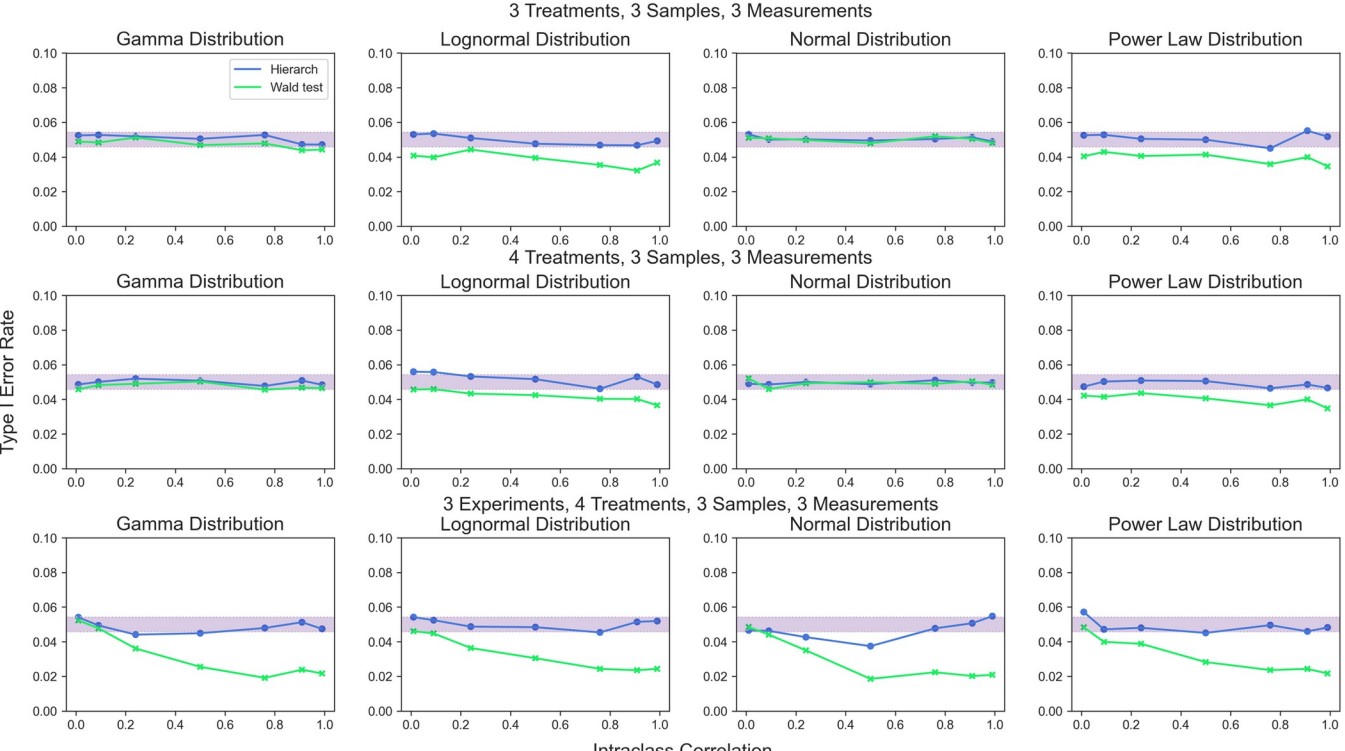

**Fig 11. Type I error rate for hierarchical randomization test and the Wald test.** Shaded area represents the 95% binomial confidence interval around a 5% type I error rate.

can produce misleading results. However, there are many situations where unbiased estimation is not necessarily desirable, and as a result, LMMs will outperform *Hierarch*, as *Hierarch* only provides unbiased estimates. However, we note that in most traditional experimental designs (split-plot, block designs, etc.), unbiased estimates are highly desirable. However, the advent of large datasets with more predictors than observations in fields such as genomics has made regularized estimation much more relevant to the life sciences. Additionally, multiple comparison correction, which *Hierarch* does provide, can be thought of as a form of regularization (though on an indicator function used to make a decision, rather than on a coefficient in a model).

In short, hierarchical randomization is a flexible strategy for performing hypothesis tests and computing confidence intervals that can handle many levels of data while maintaining robustness to non-normal errors and heteroscedasticity. Furthermore, we find the construction of the test to be pedagogical—despite our reliance on null hypothesis testing, many researchers have only a fuzzy conception of what a p value is telling them. By generating an empirical null distribution by resampling in a manner mimicking the experimental design, hierarchical permutation tests open the black box of hypothesis testing and make the meaning of a two-tailed test and the resulting p value intuitive: taking it as a given that the treatment had no effect, what are all the possible values of the test statistic that can be generated by shuffling the "treatment" labels on the data? What fraction of those possible test statistics are as or more extreme than the observed test statistic?

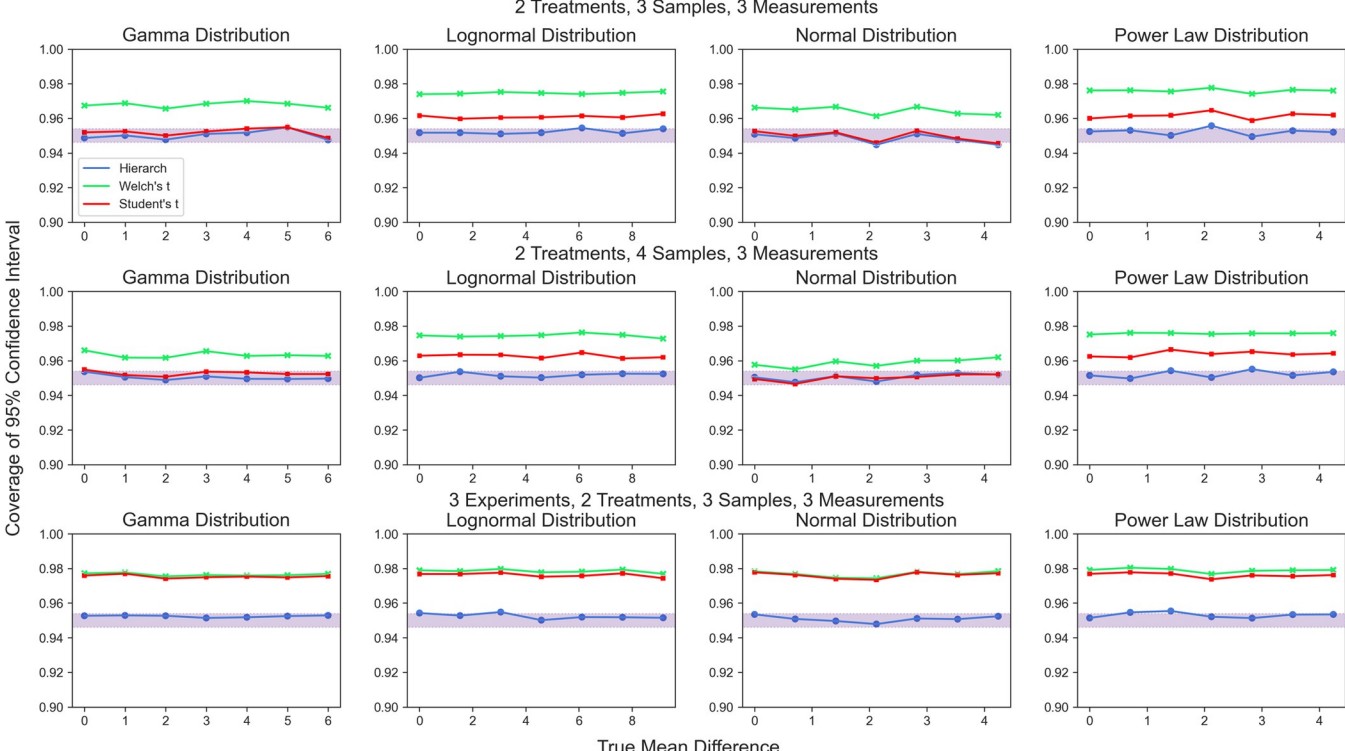

**Fig 12. Coverage probability of 95% confidence intervals generated by test inversion.** Shaded area represents the 95% binomial confidence interval around a 95% coverage probability.

## Discussion

Hierarchical randomization tests enable researchers to analyze a wide variety of experimental designs while retaining good control of Type I error rate and freedom from distributional assumptions. By using a mix of permuting and bootstrapping, these tests incorporate all of the information in a dataset without unnecessary summarization steps. In this manner, hierarchical randomization tests have an element of a Bayesian approach—rather than summarizing information from lower levels using point estimates, the randomization test is performed on the full sampling distribution of each cluster. We use simulation studies to confirm that this approach controls the two-tailed Type I error rate at 0.05 when there are as few as eight total clusters and when the intra-class correlation is sufficiently low for six clusters, which is impossible for traditional cluster permutation tests.

Despite their excellent statistical properties, randomization tests have historically been restricted to certain subfields [45–47]. We feel this is in part because setting up a randomization test often requires custom code, which results in a high computational burden and slow analyses. The software package presented in this work, *Hierarch*, makes setup and execution of a hierarchical test much simpler for practicing researchers. Not only can *Hierarch* infer the design of an experiment from the layout of the input data, but it is quite fast—every test described in this work (which use up to 10x the number of permutations necessary to compute a precise p-value) can be performed in under a second on a wide range of personal laptop computers.

We also introduce a covariance-based test statistic based on the work of DiCiccio and Romano to construct a randomization test that can be applied to linear regression problems

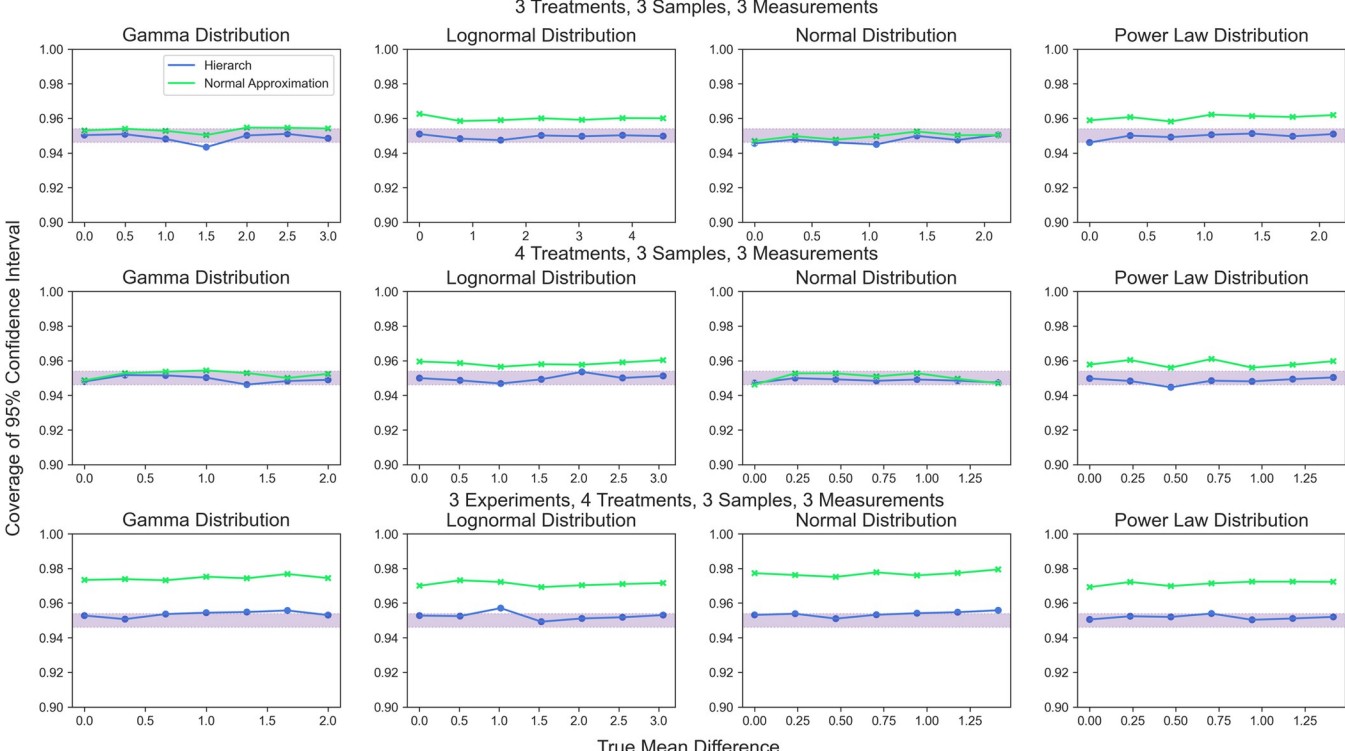

**Fig 13. Coverage probability of 95% confidence intervals generated by test inversion.** Shaded area represents the 95% binomial confidence interval around a 95% coverage probability.

[34]. This test statistic is approximately pivotal and, in the two-sample case, is linearly related to the t-statistic. We use simulation studies to confirm that this studentized covariance randomization test retains the t-statistic's Type I error rate control under homoscedasticity, and also has the desired asymptotic validity under heteroscedasticity. Notably, the test performs well even when the data is drawn from the heavily asymmetric power law distribution under heteroscedastic conditions. Even when the test fails, the Type I error rate is quite close to 5%. We demonstrate that hierarchical randomization tests based on both the t statistic and studentized covariance can be inverted to form effect size confidence intervals that maintain nominal coverage regardless of the underlying distribution.

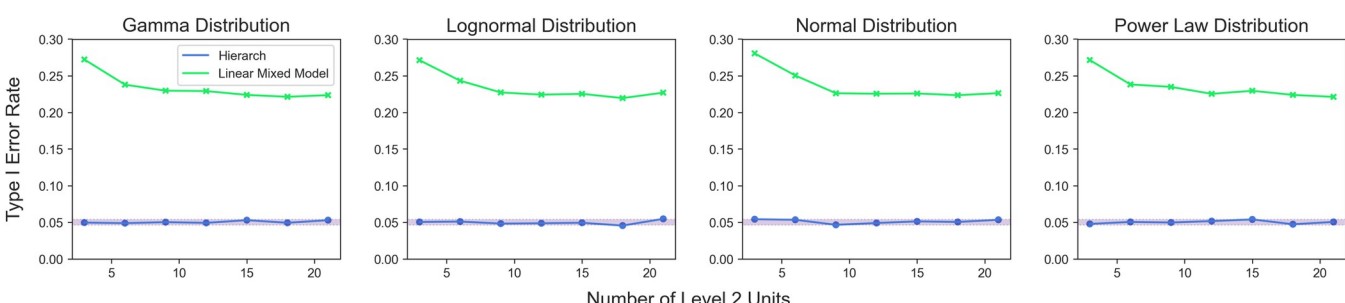

**Fig 14. Type I error rate for a hierarchical randomization test based on the studentized covariance and a linear mixed model.** Shaded area represents the 95% binomial confidence interval around a 5% type I error rate.

Hierarchical randomization hypothesis tests are versatile–they can be applied to a wide variety of experimental designs while maintaining better control of Type I error rates than asymptotic tests. While another statistical test may perform better for datasets that fulfill the assumptions of that test, these assumptions are often unverifiable–hierarchical randomization tests can be applied to any hierarchical dataset and produce an answer that does not depend on unverifiable assumptions. They do this by including multiple levels of clustering without discarding information from any level of the experimental design. These tests have good small-sample properties and are valid for several nested experimental designs common to biological research. In most cases, hypothesis testing with *Hierarch* can achieve the "platinum standard" of significance analysis without requiring researchers to produce custom code or to wait minutes or hours for their computers to produce a *p*-value.

## Supporting information

**S1 Text. The unbiased estimator for the variance of sample covariance.**
(DOCX)

**S1 Fig. Type I error rate for the hierarchical randomization test based on studentized covariance and the Wald test.**
(TIF)

**S2 Fig. Type I error rate for the hierarchical randomization test based on studentized covariance and the Wald test given unequal between-cluster variance.**
(TIF)

**S3 Fig. Type I error rate for the hierarchical randomization test based on studentized covariance and the Wald test given unequal variances.**
(TIF)

## Acknowledgments

We would like to thank Katherine Derosier, Alex Truong, and Eric Xia for helpful discussions regarding implementation of the algorithm and preparation of the manuscript.

## Author Contributions

**Conceptualization:** Rishikesh U. Kulkarni, Carolyn R. Bertozzi.

**Formal analysis:** Rishikesh U. Kulkarni.

**Funding acquisition:** Carolyn R. Bertozzi.

**Investigation:** Rishikesh U. Kulkarni, Catherine L. Wang.

**Methodology:** Rishikesh U. Kulkarni.

**Software:** Rishikesh U. Kulkarni.

**Supervision:** Carolyn R. Bertozzi.

**Validation:** Rishikesh U. Kulkarni, Catherine L. Wang.

**Visualization:** Rishikesh U. Kulkarni, Catherine L. Wang.

**Writing – original draft:** Rishikesh U. Kulkarni.

**Writing – review & editing:** Catherine L. Wang, Carolyn R. Bertozzi.

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
