## [Decision Letter · Decision Letter 0]

11 Mar 2022

Dear %TITLE% Kulkarni,

Thank you very much for submitting your manuscript "Analyzing Nested Experimental Designs - A User-Friendly Resampling Method to Determine Experimental Significance" for consideration at PLOS Computational Biology. As with all papers reviewed by the journal, your manuscript was reviewed by members of the editorial board and by several independent reviewers. The reviewers appreciated the attention to an important topic. Based on the reviews, we are likely to accept this manuscript for publication, providing that you modify the manuscript according to the review recommendations.

Sincerely,

Dina Schneidman

Software Editor

PLOS Computational Biology

[LINK]

Reviewer's Responses to Questions

**Comments to the Authors:**

Reviewer #1: The paper is very clearly written and the figures are appropriate aids to understanding the text.

At the sample sizes suggested, the number of bootstrap samples at the lowest level of the hierarchy are quite limited, and so complete enumeration rather than resampling should be used. (e.g. with 3 images per well, there are only 27 possible bootstrap means, and even with 5 there are 5^5=3125). Clearly for examples where the bottom level of the hierarchy is large (such as Example 1) resampling will be required.

The authors have demonstrated that the test statistics holds its Type 1 error rate under various scenarios and that the derived confidence intervals have the required coverage. However, they do not address power. The high coverage of the intervals for the parametric tests seems to indicate that the parametric intervals are too wide and this should be stated somewhere.

I think that referring to the bootstrap distribution as the “full posterior” is confusing. It is not the posterior – it is an estimate of the population distribution.

I would suggest not comparing the randomization test to the jackknife as the intended audience likely is not familiar with the latter.

This is a very quotable statement. Perhaps it could be highlighted in some way. “…the strong assumptions of an asymptotic test which … are doing at least as much work as the data”

Equation 7 appears to be the sample crossed moment, rather than the population crossed moment.

The code uses the order of the data columns to determine the hierarchy. It might be better to have a command line that gives the order, which would also document the hierarchy explicitly.

Is it possible to have crossed factors? E.g. Perhaps at the top level (mice) males and females, treated and untreated or a split plot design, such as treated and untreated mice and wells with different drugs? It is not clear to me if the code can handle this, and if so, how it would be indicated.

Reviewer #2: Uploaded as an attachment.

**Have the authors made all data and (if applicable) computational code underlying the findings in their manuscript fully available?**

Reviewer #1: Yes

Reviewer #2: Yes

PLOS authors have the option to publish the peer review history of their article (what does this mean?). If published, this will include your full peer review and any attached files.

Reviewer #1: No

Reviewer #2: No

Figure Files:

Data Requirements:

Reproducibility:

References:

---

## [Editor Report · Decision Letter 1]

26 Mar 2022

Dear Dr. Kulkarni,

We are pleased to inform you that your manuscript 'Analyzing Nested Experimental Designs - A User-Friendly Resampling Method to Determine Experimental Significance' has been provisionally accepted for publication in PLOS Computational Biology.

Best regards,

Dina Schneidman

Software Editor

PLOS Computational Biology

---

## [Editor Report · Acceptance letter]

26 Apr 2022

PCOMPBIOL-D-21-01262R1 

Analyzing Nested Experimental Designs - A User-Friendly Resampling Method to Determine Experimental Significance

Dear Dr Kulkarni,

I am pleased to inform you that your manuscript has been formally accepted for publication in PLOS Computational Biology. Your manuscript is now with our production department and you will be notified of the publication date in due course.

With kind regards,

Andrea Szabo
